# Celiac Disease Defined by Over-Sensitivity to Gliadin Activation and Superior Antigen Presentation of Dendritic Cells

**DOI:** 10.3390/ijms22189982

**Published:** 2021-09-15

**Authors:** Michael Hudec, Kamila Riegerová, Jan Pala, Viera Kútna, Marie Černá, Valerie Bríd O´Leary

**Affiliations:** 1Department of Medical Genetics, Third Faculty of Medicine, Charles University, Ruská 87, 100 00 Prague, Czech Republic; marie.cerna@lf3.cuni.cz (M.Č.); valerie.oleary@lf3.cuni.cz (V.B.O.); 2Department of Immunology and Clinical Biochemistry, Third Faculty of Medicine, Charles University, Ruská 87, 100 00 Prague, Czech Republic; kamila.riegerova@lf3.cuni.cz; 3Department of Pathophysiology, Third Faculty of Medicine, Charles University, Ruská 87, 100 00 Prague, Czech Republic; jan.pala@lf3.cuni.cz; 4Department of Experimental Neurobiology, National Institute of Mental Health, Topolova 748, 250 67 Klecany, Czech Republic; viera.kutna@nudz.cz

**Keywords:** monocyte-derived dendritic cells, major histocompatibility complex II, autoimmunity, CD33, CD64, CD86, MHCDQ, monocyte

## Abstract

The autoimmune condition, Celiac Disease (CeD), displays broad clinical symptoms due to gluten exposure. Its genetic association with DQ variants in the human leukocyte antigen (HLA) system has been recognised. Monocyte-derived mature dendritic cells (MoDCs) present gluten peptides through HLA-DQ and co-stimulatory molecules to T lymphocytes, eliciting a cytokine-rich microenvironment. Having access to CeD associated families prevalent in the Czech Republic, this study utilised an in vitro model to investigate their differential monocyte profile. The higher monocyte yields isolated from PBMCs of CeD patients versus control individuals also reflected the greater proportion of dendritic cells derived from these sources following lipopolysaccharide (LPS)/ peptic-tryptic-gliadin (PTG) fragment stimulation. Cell surface markers of CeD monocytes and MoDCs were subsequently profiled. This foremost study identified a novel bio-profile characterised by elevated CD64 and reduced CD33 levels, unique to CD14++ monocytes of CeD patients. Normalisation to LPS stimulation revealed the increased sensitivity of CeD-MoDCs to PTG, as shown by CD86 and HLA-DQ flow cytometric readouts. Enhanced CD86 and HLA-DQ expression in CeD-MoDCs were revealed by confocal microscopy. Analysis highlighted their dominance at the CeD-MoDC membrane in comparison to controls, reflective of superior antigen presentation ability. In conclusion, this investigative study deciphered the monocytes and MoDCs of CeD patients with the identification of a novel bio-profile marker of potential diagnostic value for clinical interpretation. Herein, the characterisation of CD86 and HLA-DQ as activators to stimulants, along with robust membrane assembly reflective of efficient antigen presentation, offers CeD targeted therapeutic avenues worth further exploration.

## 1. Introduction

Celiac disease (CeD) is a chronic autoimmune condition that affects 1–2% of the general population, ensuing in genetically predisposed individuals that harbour variants of the human leukocyte antigen (HLA) system [1]. HLA class II are alpha (α)–beta (β) heterodimeric cell surface receptors found on antigen-presenting cells. The α and β chains are encoded by two adjacent loci, *HLA–DQA1* and *HLA–DQB1*, respectively, on chromosome 6p21.3 [2]. A serotyping system classifies the variant *HLA-DQ2* (alleles DQA1 *05:01/DQB1 *02:01) and *HLA-DQ8* (alleles DQA1 *03:01/DQB1 *03:02) [3]. These cis-haplotype isoforms are directly associated with CeD [3,4].

Primarily affecting the small intestine, CeD manifests as a spectrum of broad clinical symptoms including fatigue, weight loss, anaemia, nausea, constipation or diarrhoea [5]. Pathological alterations include villous atrophy, crypt hyperplasia, inflammatory cell infiltration and activation [6]. CeD can be triggered by gluten, a combination of prolavin and glutelin proteins, which naturally occurs in wheat, barley, rye, oats and their cross hybrids [7]. Upon ingestion, gluten is incompletely cleaved into short peptides by gastric, pancreatic and brush border enzymes [8] that traverse the lamina propria of the small intestine via trans- or paracellular routes, activating a CeD autoimmune response [9,10,11]. Such gluten peptides are presented by monocyte-derived mature dendritic cells (DCs), through HLA-DQ and co-stimulatory molecules CD40, CD80 and CD86 to CD4 positive T lymphocytes, eliciting a cytokine- and chemokine-rich microenvironment accompanied by p38 MAP kinase phosphorylation [12,13]. There are changes in the proportional representation of monocytes, particularly intermediate or non-classical subpopulations, in autoimmune diseases [14,15]. Intermediate monocytes are able to overexpress HLA-II molecules [16] and, in common with the non-classical type, may play a role in CD8 cellular activation [17,18]. To date, limited knowledge exists on the differential monocyte profile in the context of the CeD immune response. Likewise, the differentiation of monocytic to dendritic cells needs to be deciphered in the context of HLA-DQ involvement within CeD cellular responses to gluten-derived peptides. Enhanced cytokine production from HLA-DQ2+ monocytes following gliadin stimulation has been noted in patients with active CeD compared with monocytes derived from healthy donors [19]. This work interpreted the comparative changes in monocytes and their derived dendritic cells in CeD individuals with a familial autoimmunity association against normal controls within the same pedigree background. Furthermore, this investigation elucidated the role of cell surface receptors such as CD86 and HLA-DQ expression profiles in monocyte-derived dendritic cells as potential controllers of gluten induction, leveraging CeD adaptive immunity towards self- targeting.

## 2. Results

### 2.1. High Prevalence of CeD in the Czech Republic with Study Participants Representing Autoimmune Disease Susceptibility across Familial Generations

According to the World Gastroenterology Organisation (WGO) Global 2016 guidelines [20], the Czech Republic is amongst the European countries with the highest prevalence of CeD (Figure 1A). While the highest incidence of CeD is found in Finland (2%), the percentage prevalence of CeD in the Czech Republic is 0.3% (1 in 218 individuals) [20]. Three Czech families (F-A, F-B, F-C) with a history of autoimmune disease participated in this study. Healthy controls within these families were also enrolled. Their pedigree overview is presented in Figure 1B, with CeD only affecting females arising in the second generation in F-B and F-C. Third generation females of F-A were affected by a combination of CeD and autoimmune inflammation of the thyroid gland (Figure 1B). Members of F-A, F-B and F-C had allergic reactions to many factors, especially pollen and dust mites (Figure 1C, Appendix A). From the available data, it was shown that CeD manifested by the second decade of life, with the earliest reported by age 5 years in a female from F-A (Figure 1D). Two participants elicited antigen positivity towards thyroid peroxidase antibody (TPO-ab) (Figure 1D). Six members across the three families had autoimmune inflammation of the thyroid gland. Members of these representative families donated peripheral whole blood with no statistically significant difference between generations in the proportion of monocytes isolated from this source (Appendix A).

### 2.2. Differential Expression of Transmembrane Markers Indicates Monocytes Extracted from CeD Patients Have the Ability to Transform into Dendritic Cells after Exposure to LPS/PTG

Monocytes, the largest leukocyte type, are spherical in shape (Figure 2A upper) with prominent surface protrusions upon differentiation with LPS/PTG to dendritic cells (Figure 2A lower). The frequencies of lymphocytes, monocytes and dendritic cells in PBMCs from healthy individuals were expected to be in the range of 70–90%, 10–20% and 1–2%, respectively [21]. Monocytes isolated from PBMCs from CeD individuals were almost within the published range (9 ± 3%; Figure 2B,C upper), while healthy controls (CTL) were slightly outside it (5 ± 1%; Figure 2C upper). However, differences in monocyte yield arose between Group 1 and 3 (Figure 2C upper) as seen when CTL were compared to CeD (*p* = 0.0357) as determined by size and granularity (forward versus side scatter) flow cytometric plot distribution profiles. Individuals with thyroid hormone imbalance (Group 2) showed a 4.5 ± 1% monocyte yield, which differed significantly from CeD (*p* = 0.028) but not when compared to CTLs (*p* = 0.267) (data not shown). Gating strategies indicated monocytes have the ability to transform into MoDCs in CTL and especially CeD groups (41 ± 3%; Figure 2C lower, Appendix A) after exposure to LPS/PTG for 1 day. Monocytes derived from CeD patients had a significantly higher ability to transform into MoDCs when compared to the CTLs, with an approx. 4.2-fold increase in percentage proportion of MoDCs (*p* = 0.00083, Figure 2C lower). Likewise, the percentage proportion of CeD MoDCs differed significantly from individuals with thyroid hormone imbalance (6.5-fold increase, *p* = 0.011), which showed similar levels to CTLs (*p* = 0.412, data not shown).

Monocytes were characterised as classical, intermediate and non-classical via profiling of surface marker expression (Figure 2D upper). Results showed that 80.26 ± 3% were CD14++ CD16− classical monocytes in CeD patients. In this patient group, 4.28 ± 2% were CD16+ CD14++ intermediate monocytes and 10.31 ± 1% CD16++ CD14+ non-classical monocytes, while 5.15 ± 0.5% of cells were negative for both CD16 and CD14 and as such, were not characterised. No statistically significant difference was observed between CTL and CeD groups when monocyte subpopulations were compared (classical monocytes CD14++CD16− (*p* = 0.4121), intermediate CD14++CD16+ (*p* = 0.3152) and non-classical CD14-CD16++ (*p* = 0.3152)). It should be noted that monocyte subsets are not homogeneous populations with underlying distinct transcriptional cellular profiles [22]. While the emphasis was placed on CD14++ expressing monocytes given their prominence in this cell type, classical, intermediate and non-classical categories were not deliberately separated prior to MoDC transformation for this study.

MoDCs from CTL and CeD were predominantly CD11c+CD86+ according to gating demarcation (91 ± 4%) (Figure 2D lower). Surface marker CD14, CD86, CD11c and HLA-DQ profiling altered significantly during the transformation of CeD monocytes to MoDCs, as can be seen in the forward scatter patterns (Figure 2E,F). Mean fluorescence intensity values for CD14 were 15-fold higher in CeD monocytes compared to MoDCs (*p* = 0.00001). In contrast, CD86, CD11c and HLA-DQ levels were 56, 30 and 428-fold higher, respectively, in MoDCs in comparison to CeD monocytes (CD86, *p* = 0.00024; CD11c, *p* = 0.00055; HLA-DQ, *p* = 0.00133).

### 2.3. Identification of a Novel CD33:CD64 Bio-Profile Marker for CeD Patients

The transmembrane receptor CD33 is expressed on cells derived from the bone marrow of myeloid lineage [23]. CD64 is an integral membrane glycoprotein and is constitutively only found on monocytes and macrophages [24]. Following selection for CD14++ expression (see above), monocytes from CeD patients were analysed for their expression of CD33 and CD64. There is no statistically significant difference in the expression of CD33 (*p* = 0.4020) and CD64 (*p* = 0.3650) in CD16+ monocytes in CTLs. However, findings revealed that CD14++ monocytes from CeD patients have significantly higher levels of CD64 expression (*p* = 0.0352) than CTL along with lower CD33 expression when similarly compared (*p* = 0.0029). While CD33 and CD64 had similar levels in CTL (approx. 10^4^/mL blood), intriguingly, CD33 and CD64 showed a highly significant inverse profile in this study (*p* = 0.00012; Figure 3A right). This finding highlights a novel CeD bio-profile marker that could be exploited for diagnostic purposes.

### 2.4. MoDCs from CeD Patients Exhibit Hypersensitivity to Gliadin Activation

The type I transmembrane integrin alpha chain protein, CD11c, is found at high levels on dendritic cells [25]. This marker was chosen as a positive selection marker of CeD or CTL monocyte differentiation. Likewise, CD86 acted as an indicator of MoDC maturation [26]. In response to LPS/PTG stimulation, CD11c+ MoDCs originating from CeD patients revealed a higher proportion of CD86 expression in comparison to CTL samples (Figure 3B and Appendix A). Significant fold differences were noted when levels were relatively compared against a normalised LPS background (0.7-fold increase, *p* = 0.0476; Figure 3B right). CD11c+ MoDCs were further assessed for their expression of the HLA-DQ surface marker after LPS or PTG. Similar HLA-DQ stimulation profiles were found following exposure to LPS or PTG in CD11c+ MoDCs originating from CeD patients (*p* = 0.625; Figure 3C and Appendix A). This contrasted with the lower HLA-DQ levels in CD11c+ MoDCs originating from CTL when exposed to PTG fragments in comparison to LPS stimulation (*p* = 0.046; Figure 3C middle). When compared as indicated above, significantly increased HLA-DQ fold differences were revealed in CD11c+ MoDCs originating from CeD patients versus CTL when levels were relatively compared against a normalised LPS background (2.2-fold increase, *p* = 0.0179; Figure 3C right). Elevated CD86 and HLA-DQ expression in CD11c+ MoDCs to PTG exposure highlights the hypersensitivity of CeD patients to this stimulant in contrast to the general homeostatic response.

### 2.5. Superior CD86 Presentation in MoDCs from CeD Patients Compared to CTL

The intracellular distribution of the MoDC maturation indicator CD86 was sought, given the evidence reported above that CeD is characterised by its elevated levels upon gliadin fragment exposure. Confocal microscopic analysis revealed the predominance of CD86 accumulating within the MoDC cellular membrane in CeD samples (Figure 4A) in comparison to CTLs (Figure 4B). Results showed a very significant increase (0.6-fold elevation; *p* = 0.0037, Figure 4C,D) in fluorescence intensity signal corresponding to CD86 at the membrane in CeD samples. In contrast, CD86 appeared to be confined to the cytosol in MoDCs originating from healthy control individuals (Figure 4B). Similar CD86 cytosolic expression levels (*p* = 0.381) were noted for both CeD and CTL groups (Figure 4E). This supports previous findings with diabetic patients highlighting that mature MoDCs upregulate the synthesis of co-stimulatory receptor molecules, such as CD86, during antigen presentation [26].

### 2.6. Predominant Membrane-Associated HLA-DQ Expression in MoDCs Originating from CeD Patients Compared to CTL

HLA-DQ is a cell surface receptor found on antigen-presenting cells [3]. High levels of HLA-DQ2 and HLA-DQ8 alleles have assisted in identifying CeD patients and individuals at risk of developing this condition [27]. Confocal micrograph assessment of MoDCs from CeD patients strongly supported the sensitivity of HLA-DQ recognition to 24 h PTG exposure, in contrast to those from CTL individuals. Robust expression of HLA-DQ predominated at the MoDC membrane in CeD samples as seen from the highly significant elevated mean fluorescence intensity profiling in comparison to controls (Figure 5A–C). HLA-DQ revealed a 2.2-fold higher level of membrane associated expression (*p* = 0.00028) in CeD-MoDCs versus CTL-MoDCs (Figure 5D). Furthermore, the presence of HLA- DQ was also very much increased in the cytosol of CeD-MoDCs compared to CTL-MoDCs (1.8-fold upregulation, *p* = 0.0001; Figure 5E). Such findings agree with the known functional characteristics of HLA-DQ receptors [28] in antigen recognition and presentation to CD4+ T helper-cells to enable activation of the host innate immunity system.

## 3. Discussion

This investigation deciphered the cell surface profile of monocytes and their derived dendritic cells in the context of CeD. Focus was placed on the involvement of HLA-DQ and co-stimulatory surface membrane markers due to their recognised association with autoimmunity [29]. The prevalence of CeD in the Czech Republic enabled access to PBMC for in vitro model utilisation from three families with a history of autoimmune disease. Differential CD14 CD16 positivity facilitated monocyte identification. Findings revealed substantially greater monocyte yields from CeD patients compared to healthy controls. Further analysis identified a novel bio- profile characterised by elevated CD64 and reduced CD33 levels, unique to the CD14++ monocytes of CeD patients. The greater sensitivity of CeD monocytes to LPS-PTG stimulation was initially seen from the higher proportions of dendritic cells from this source. Elevated CD86, CD11c and HLA-DQ cell surface markers confirmed the efficient transformation of monocytes into mature dendritic cells and empowered the effectiveness of the in vitro platform as a CeD model system. In comparison to controls, an increased sensitivity to PTG from CD86 and HLA-DQ readouts was found in MoDCs originating from CeD patients. Delving further into the influence of LPS-PTG stimulation on MoDCs, confocal microscopy demonstrated that the intracellular distribution of CD86 and HLA-DQ predominated at the cellular membrane in CeD, indicative of superior antigen presentation ability.

Despite the advances in CeD detection, its prevalence of 1% within the European Union has only recently been published using a large population cohort [30]. Findings have shown that CeD varies considerably in different European countries, with Finland having the highest prevalence at approx. 2% [30,31]. Recent data estimated the occurrence of CeD in the Czech Republic at 0.3% [20], with others reporting an even greater increase in the condition in that country [32]. The reason for the rise in CeD occurrence is unknown, although environmental factors linked to hygiene have been suggested [33]. Nevertheless, first-degree relatives (FDR) of individuals with CeD form an especially high-risk group, estimated at 4–17% [34,35]. For FDR homozygous for HLA-DQ2, the chances of developing CeD during early age reaches 26% [36], with second-degree relatives also at increased risk levels, although to a lesser extent [37].

It is acknowledged that current CeD in vitro model systems are unable to recapitulate the multifactorial aspects of this complex disorder [38]. Nevertheless, the recently developed strategy for exploiting peripheral blood mononuclear cells that harbour B cells, gluten-reactive T cells, dendritic cells and monocytes offers a valuable avenue for establishing a hierarchy of immunogenic peptides involved in the immune response of CeD individuals. Advantage can be taken of the limited diversity of the pathogenic epitopes as a basis for using gliadin-digested peptide fragments within the CeD in vitro model system. Focus has been placed by others on T-cell characterisation from PBMCs with pMHCII tetramers [39]. This technique has identified upregulated CD38 on T cells as a possible biomarker for gluten re-exposure or diagnosis of CeD [40]. Nevertheless, the pMHCII tetramer technique still needs additional PBMC techniques to support its performance [39]. Herein, this study isolated PBMCs from CeD and CTL subjects and focused instead on CD14++ monocytes and their subsequent transformation to mature monocyte (Mo)-derived dendritic cells (DCs). Flow cytometric profiling and mean fluorescence intensity analysis enabled accurate selection of CD14++, CD16 monocytes originating from CeD subjects and CTLs. The efficacy of monocyte transformation to MoDCs upon LPS/PTG stimulation was evident in selective upregulation of CD86, CD11c and HLA- DQ maturation indicator markers. Importantly, this study identified a novel CeD bio-profile represented by the inverse relationship between CD33 and CD64 surface markers in CD14++ monocytes originating from CeD patients. Through the careful decipherment of flow cytometric readouts, reduced CD33 levels were associated with elevated levels of the co-stimulatory partner CD64. The significance of this finding needs further elucidation given that CD33 represents a lineage marker associated with phagocytotic inhibition [41,42] and CD64 is an early onset marker of clinical infection [43]. Nevertheless, it may be of potential diagnostic and therapeutic value for CeD remediation.

Over-sensitivity of MoDCs to gliadin was revealed from increased expression of CD86 following normalisation to LPS, indicative of pro-inflammatory pathway activation in CeD versus CTLs [44]. Others have noted that CD11c+ dendritic cells accumulated in the celiac lesion and revealed an activated phenotype expressing CD86 [45], considered one of the key molecules involved in the co-stimulation of T cells [46]. CeD demonstrates very strong HLA association, higher than that seen in many other auto-immune diseases such as type I diabetes or rheumatoid arthritis [47]. While the response was lower than that elicited by LPS in this CeD in vitro model, this study showed the elevated expression of HLA-DQ in MoDCs originating from CeD subjects compared to CTLs, indicative of the increased sensitivity to gliadin activation.

Confocal microscopy and analysis provided evidence demonstrating the predominance of CD86 and HLA-DQ at the surface membrane of MoDCs originating from CeD patients. Comparative mean fluorescence levels were noted for CD86 cytosolic levels for both CeD and CTL groups. Research by others noted the constitutive expression of *CD86* mRNA in mini porcine aortic endothelial cells enabling rapid translation of the CD86 protein on the cell surface following activation, presumably from the stored mRNA [46]. Recently, two alternatively spliced *CD86* mRNA variants were identified in resting human monocytes characterized by a deletion of the transmembrane domain (soluble CD86) and extracellular (EC) domain (transmembrane form of CD86), respectively [48]. The implications of the presence of soluble CD86 on CeD aetiology is currently unknown. The robust expression of CD86 at the MoDC membrane after gliadin peptide exposure provides evidence for its role in superior antigen presentation in CeD. In contrast HLA-DQ levels were strongly elevated in both cytosolic and membrane compartments of MoDCs from CeD versus CTLs. Such data supports the specific association of HLA-DQ with CeD and its recognised presentation ability.

In conclusion, this investigative study interpreted the monocytes and MoDCs of CeD patients with the identification of a novel bio-profile marker for clinical interpretation. Herein, evidence has shown that CD86 and HLA-DQ are reactors to CeD activation, triggering their congregation at the MoDC membrane for efficient antigen presentation. Furthermore, harnessing the attributes of the in vitro model with inclusion of highly potent MoDCs will offer robust therapeutic avenues for CeD patients and their future family pedigree.

## 4. Materials and Methods

### 4.1. Clinical Diagnosis of CeD Patients and Pedigree Overview

CeD was diagnosed in this study population following current diagnosis guideline, i.e., measurement of total serum IgA and IgA antibodies against transglutaminase 2 (TGA-IgA). If TGA-IgA was 2–10 times the upper limit of normal (10× ULN), biopsy diagnosis was applied [49]. Written informed consent was obtained from all individuals involved in this study with approval from the Ethics Committee of the Institute for Clinical and Experimental Medicine and Thomayer Hospital, Prague, Czech Republic (approval number G-18–23). A total of 15 participants (*n* = 15) were enrolled in this study and categorised as follows: gluten-free diet (GFD)-treated CeD patients (Group 1: *n* = 4); autoimmune disease patients (Group 2: *n* = 4); normal controls with no known autoimmune disease, inflammation or malignancy (Group 3: *n*= 7). Detailed patient characteristics can be found in Figure 1D. All subjects were Caucasians of European ancestry. In terms of pedigree, Group 1–3 belonged to three families (F-A, F-B and F-C) from the Czech Republic with a history of autoimmune disease. The familial cohorts were structured as follows: Members–7 (F-A), 4 (F-B, F-C); Generations– 3 (F-A), 2 (F-B, F-C); and Distinct autoimmune diseases–2 (F-A, F-B, F-C).

### 4.2. Monocyte Isolation from Peripheral Blood Mononuclear Cells

Peripheral whole blood was acquired from participants (outlined above) by a phlebotomist based in the Department of Medical Genetics of Third Faculty of Medicine, Charles University, Prague, CZ. Following dilution with RPM1 1640 medium (1:1; Sigma-Aldrich, St. Louis, MO, USA), whole blood was gently layered over Ficoll-Paque in a 1:0.7 ratio (GE HealthCare Bio-Sciences AB, Uppsala, Sweden) in a Falcon tube and centrifuged for 30 min at 400× *g*, resulting in the formation of distinct layers. Peripheral blood mononuclear cells (PBMCs) were located at the interface of the top and second layers beneath the plasma zone. The plasma layer was removed by pipetting, and PBMCs were transferred to room temperature (RT) complete RPM1 1640 medium (Sigma- Aldrich, MO, USA) to remove any remaining platelets. Following centrifugation for 20 min at 400× *g*, samples of PBMCs were stained with trypan blue, viewed under a light microscope and counted on a haemocytometer slide. PBMCs (10^6^ cells) were cultivated for 2 h in a 75 cm^2^ plastic culture flask (Schoeller Pharma, Prague, Czech Republic) in complete medium (CM) containing RPMI 1640, foetal bovine serum (10%; Sigma-Aldrich, MO, USA) in the presence of antibiotic-antimycotic solution (1%; Sigma-Aldrich, MO, USA) at 37 °C in 5% CO_2_ atmosphere. The non-adherent fraction was then removed following washing with CM.

### 4.3. Generation of Monocyte Derived Dendritic Cells (MoDCs)

Adherent monocytes were cultivated in CM1 (CM with inclusion of granulocyte-macrophage colony-stimulating factor human (GM-CSF) (500 ng/mL; Sigma-Aldrich, MO, USA) and human interleukin-4 (IL-4) (200 ng/mL; Sigma-Aldrich, MO, USA) for 6 days. Generated MoDCs (CD11c^+^, CD14-) were harvested manually with a cell scraper, centrifuged for 5 min at 400× *g*, resuspended in CM1 and divided into 6-well plates (Schoeller Pharma, Prague, Czech Republic). Gliadin (Sigma-Aldrich, MO, USA) was fractionated by pepsin-trypsin digestion as follows: gliadin (100 mg/mL) and pepsin (2 mg/mL; Sigma-Aldrich, MO, USA) were dissolved in 0.2 N HCl (Sigma-Aldrich, MO, USA) and incubated for 2 h at 37 °C with pH adjustment to 7.4 before addition of trypsin (2 mg/mL). The solution was incubated for 4 h at 37 °C with rotation at 250 rpm. The enzymatic reaction was stopped by incubation for 30 min at 100 °C. Peptic-tryptic-gliadin (PTG) fragments were divided into aliquots and frozen at −20 °C. PTG (1 mg/mL) and lipopolysaccharides (LPS) (100 ng/mL; Sigma-Aldrich, MO, USA) were added to cells for 24 h to induce MoDC maturation.

### 4.4. Flow Cytometry and FACS Staining

After 24 h stimulation by LPS/PTG, MoDCs were manually removed using a cell scraper, briefly centrifuged at 400× *g* and resuspended in 100 μL RPMI 1640 medium (Sigma-Aldrich, MO, USA) with monoclonal antibodies (see below) added according to manufacturer’s recommendation, and incubated for 20 min at 4 °C in the dark. The following anti-human monoclonal antibodies (EXBIO, Prague, Czech Republic) were used: PE-conjugated CD86 (clone BU63), FITC-conjugated CD14 (clone MEM-15), PerCP-conjugated CD16 (clone 3G8), APC-conjugated CD33 (clone WM53), PerCP-conjugated CD11c (clone BU15), PE-conjugated CD64 (clone 10.1) and APC-conjugated HLA-DQ (clone SK10, Thermo-Fisher Scientific, Waltham, MA, USA). An additional wash was performed after incubation. MoDCs were resuspended in cold PBS (Sigma-Aldrich, MO, USA) with analysis performed on a BD FACSVerse flow cytometer (BD Biosciences, San Jose, CA, USA). Data were analysed with BD FACSuite™ Software. All images and graphs were generated and assembled using GraphPad PRISM v3.0 and Inkscape v1.0.1.

### 4.5. Immunohistochemical and Confocal Microscopy Detection of MoDC Surface Antigens

MoDCs were cultivated as outlined above on glass coverslips. After 24 h stimulation by LPS/PTG, growth medium was removed and MoDCs were 2 times for 5 min with PBS. These cells were exposed to 4% paraformaldehyde for 15 min by way of fixation, followed by washing for 5 min with PBS. Cells were then permeabilized in 1× TBST (1× TBS including 0.5% Triton™ X-100 (Sigma-Aldrich MO, USA)) for 1 hr. Following one wash for 5 min in 1× TBS, MoDCs were placed in blocking solution (1× TBS containing 5% bovine serum albumin and 0.5% Triton^TM^ X-100) for 1 h at RT. These cells were then exposed to PE-conjugated anti-CD86 (1:100 in blocking solution; details above) or APC-conjugated anti- HLA-DQ (1:100 in blocking solution; details above) with 2 h incubation at 4 °C. Samples were washed 3 times for 15 min in 1× TBST and air-dried in the dark. To prepare for microscopy, MoDCs grown on coverslips were mounted in ProLong^®^ Gold Antifade reagent (Cell Signalling Technology, Beverly, MA, USA; cat. # 8961S) containing DAPI and placed on glass slides. Microscopic images were acquired as single XY planes with the pinhole set to Airy1 on an inverted confocal laser scanning microscope Leica TCS SP5 (DMI6000, Leica Microsystems, Mannheim, Germany) using an HCX PL APO 30×/1.30 Oil objective. Images were analysed in LAS AF software (Leica Microsystems, Mannheim, Germany) and ImageJ v1.53e software (NIH, Washington, DC, USA). All images and graphs were generated and assembled in figures using GraphPad PRISM v3.0 and Inkscape v1.0.1.

### 4.6. Statistical Analysis

Data comparison was performed using nonparametric, one-tailed Mann–Whitney U test, two-tailed Wilcoxon matched-pairs signed rank test or Student’s *t*-test using GraphPad PRISM 3.0 or Microsoft Excel. Values of *p* ≤ 0.05 were considered significant.

## Figures and Tables

**Figure 1 ijms-22-09982-f001:**
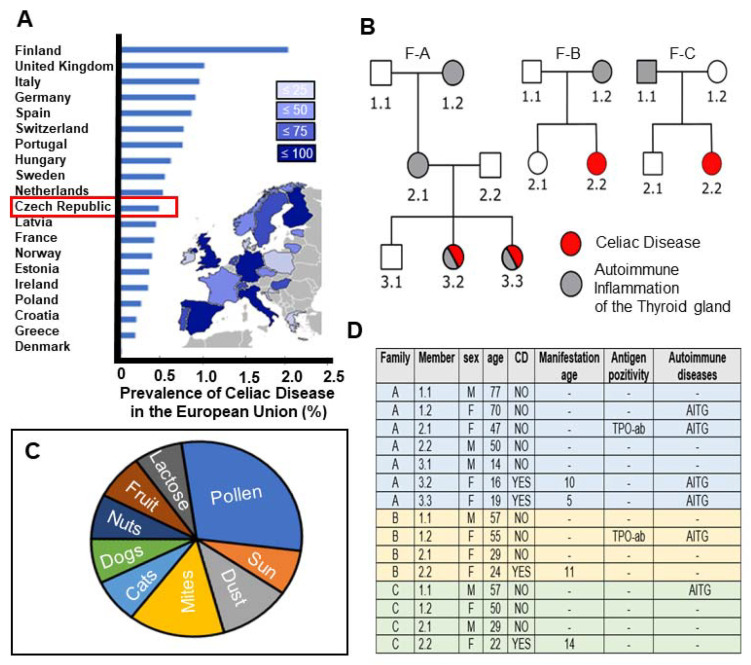
High prevalence of CeD in the Czech Republic with study participants representing auto immune disease susceptibility across familial generations. (**A**) Horizontal histograms representing the percentage prevalence of CeD in various European countries. Data sourced from WGO (https://www.worldgastroenterology.org/guidelines, accessed on 12 June 2021) with modifications [20]. Inset map of European countries showing the occurrence of CeD per their respective total populations. (**B**) Pedigree overview of three families (left: F-A; middle: F-B; right: F-C) whose members participated in this study. Individuals affected by CeD (red), autoimmune inflammation of the thyroid gland (grey), both CeD and autoimmune inflammation of the thyroid gland (red-grey) or non-affected controls (white). (**C**) Pie chart depicting the types of allergens afflicting the study participants. (**D**) Table summarising the sex, age, CeD occurrence and manifestation age in family members participating in this study. Antigen positivity to gliadin or autoimmune inflammation of the thyroid gland (AITG) in some family members were indicated. Two participants elicited antigen positivity towards thyroid peroxidase antibody (TPO-ab).

**Figure 2 ijms-22-09982-f002:**
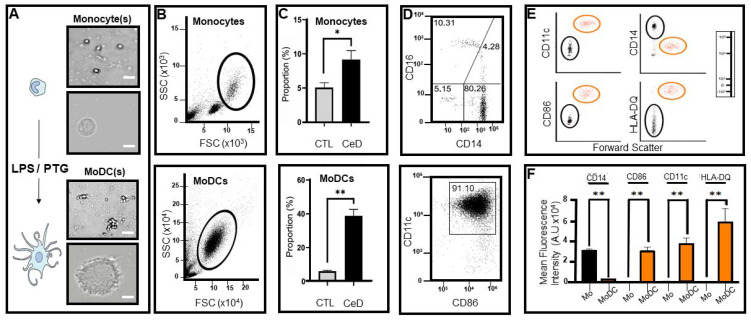
Differential expression of transmembrane markers indicates monocytes extracted from CeD patients have the ability to transform into dendritic cells after exposure to gliadin and lipopolysaccharide. (**A**) Representative schematic (left) and light microscope images of monocytes (right, upper) extracted from CeD peripheral mononuclear cells (PBMCs). Following exposure to lipopolysaccharide (LPS) or peptic-tryptic-gliadin (PTG) fragments, monocytes differentiated into monocyte-derived dendritic cells (MoDCs; Schematic (left), representative images (right, lower)). Images taken at 10× magnification; scale bar 100 μm. Inset images acquired at 40× magnification; scale bar 25 μm. (**B**) Dot plots indicating side scatter (SSC) versus forward scatter (FSC) profiles for CeD monocytes (upper) or MoDCs (lower). Gating demarcates the percentage recovery of monocytes from peripheral mononuclear cells (upper) or transformation efficiency to MoDCs post 24 h exposure to LPS/PTG (lower). (**C**) Histograms showing the significant difference in the proportional percentage of monocytes extracted from PBMCs (upper) and transformed MoDCs (lower) from healthy controls (CTL) and celiac disease patients (CeD). Asterisk (*) or asterisks (**) represent statistical significance at *p* < 0.05 or *p* < 0.005 respectively. (**D**) Representative dot plot showing predominance of CD14++, CD16- monocytes in CeD PBMCs (upper) or CD11c+, CD86+ MoDCs (lower). Numbers indicate percentage of CD16+ and/or CD14+ surface markers on monocytes (upper) or CD11c+, CD86+ MoDCs (lower), respectively. (**E**) Forward scatter dot plots of monocytes (black gate) or MoDCs (orange gate) differentially expressing surface markers CD14, CD86, CD11c or HLA-DQ. Logarithmic scale provided (right). (**F**) Histograms showing significant difference in the mean fluorescence intensity for CD14, CD86, CD11c or HLA-DQ between monocytes (Mo) and MoDCs. Asterisks (**) represent statistical significance at *p* < 0.005.

**Figure 3 ijms-22-09982-f003:**
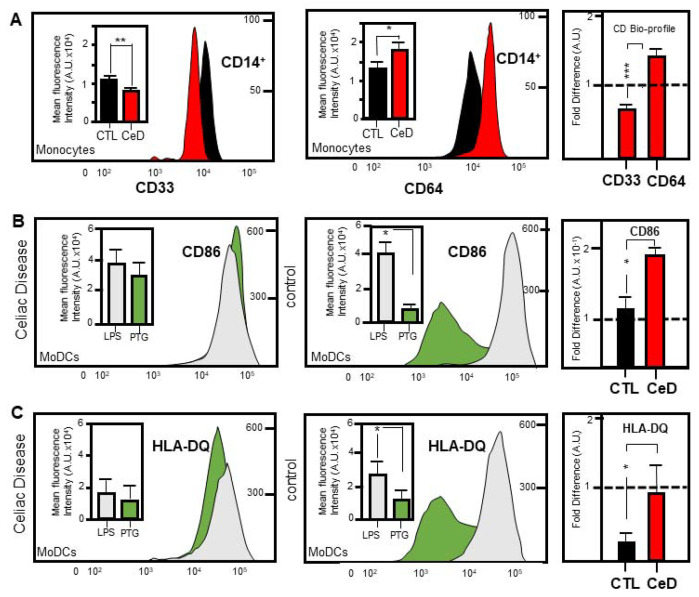
CeD defined by the identification of a novel bio-profile marker CD33 CD64++ and hyper-sensitivity to gliadin activation. (**A**) Curved flow cytometric histograms (CFCH) reveal the lower or higher expression of CD14++ monocyte surface markers CD33 (left) or CD64 (middle), respectively, in CeD patients versus healthy controls (CTL). Insets represent the mean fluorescence intensities as arbitrary units (A.U.) for the combined CD33: CD14++ (left) or CD64: CD14++ profile for CTL versus CeDs. Percentage scale bar shown. Histograms represent the highly significant fold difference between CD33 and CD64 levels in CeD patients when normalised to those found in CTL (dashed line). Asterisks (*, ** or ***) represent statistical significance at *p* < 0.05, *p* < 0.005 or *p* < 0.0005 respectively. (**B**,**C**) CFCH and insets show expression of MoDC surface marker CD86 (**B**) or HLA-DQ (**C**) to gliadin (green) compared to LPS (grey) in CeD (left) versus CTL (middle). Scale represents cellular population levels or mean fluorescence intensity A.U. Histograms underscore the highly significant increased sensitivity of CeD (red) versus CTL (black) to gliadin activation (right). Asterisk (*) represent statistical significance at *p* < 0.05.

**Figure 4 ijms-22-09982-f004:**
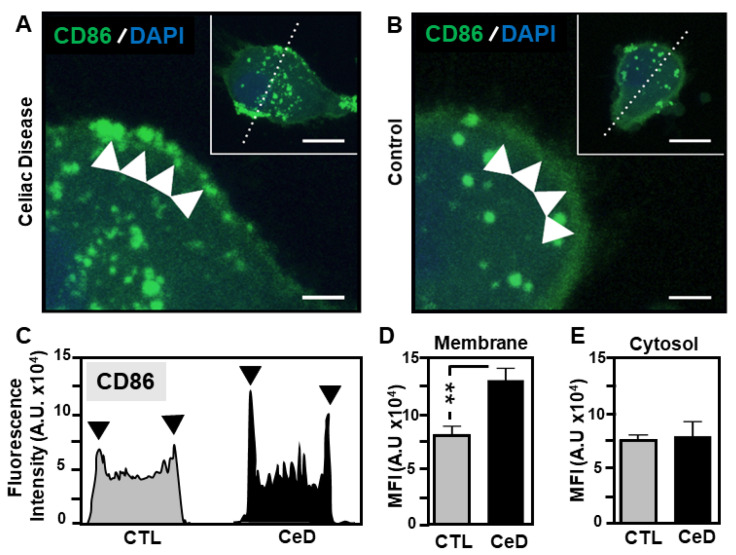
Superior CD86 presentation in MoDCs from CeD patients compared to CTL. (**A**,**B**) Representative confocal micrographs of CD86 expression in MoDCs originating from CeD (CeD–MoDCs) patients (**A**) or CTL (**B**). Note the predominance of CD86 localised to the surface of CeD-MoDCs compared to CTL samples (green, white arrowheads). Scale bar 10 μm. Inset showing an overview of an individual MoDC transected (dashed line) for fluorescence intensity profile determination. Scale bar 5 μm. Nucleus stained with DAPI (blue). (**C**) Fluorescence intensity profile of MoDCs transected in insets (A and B). Black arrowheads indicate CD86 detected at the MoDC surface membranes of CeD and control (CTL) samples. (**D**,**E**) Histograms showing the mean fluorescence intensity (MFI) as arbitrary units for CD86 in the membrane (**D**) or cytosolic (**E**) regions of MoDCs from CeD or CTLs. Asterisks (**) represent statistical significance at *p* < 0.005.

**Figure 5 ijms-22-09982-f005:**
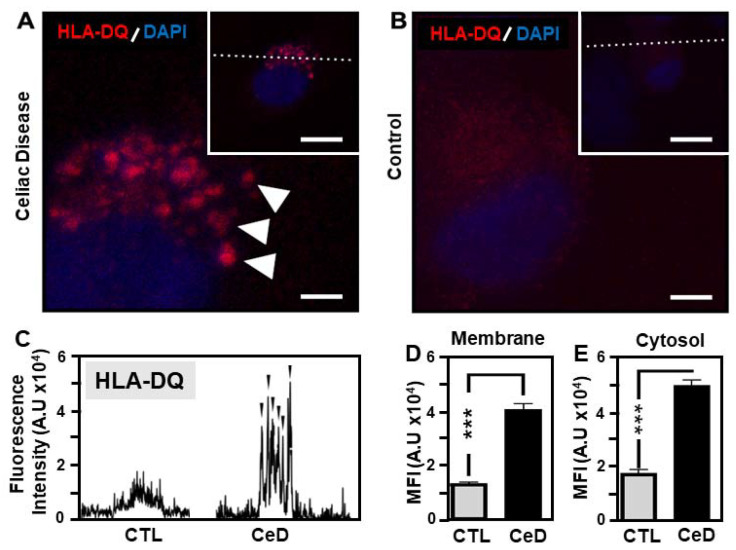
Robust HLA-DQ membrane expression in MoDCs from CeD patients compared to CTL. (**A**,**B**) Representative confocal micrographs of HLA-DQ expression in MoDCs originating from CeD (CeD–MoDCs) patients (**A**) or CTL (**B**). Note the high expression levels of HLA-DQ localised to the surface of CeD-MoDCs compared to CTL samples (red, white arrows). Scale bar 5 μm. Inset showing an overview of an individual MoDC transected (dashed line) for fluorescence intensity profile determination. Scale bar 20 μm. Nucleus stained with DAPI (blue). (**C**) Fluorescence intensity profile of MoDCs transected in insets (**A**,**B**). Black arrowheads indicate HLA-DQ detected at the MoDC surface membranes of CeD and CTL samples. (**D**,**E**) Histograms showing the mean fluorescence intensity (MFI) as arbitrary units for HLA-DQ in the membrane (**D**) or cytosol (**E**) of MoDCs from CTL or CeD. Asterisks (***) represent statistical significance at *p* < 0.0005.

## Data Availability

The data presented in this study are available on request from the corresponding author. The data are not publicly available due to ethical restrictions.

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
