# Peer review of "Celiac Disease Defined by Over-Sensitivity to Gliadin Activation and Superior Antigen Presentation of Dendritic Cells"

_ijms, 2021, doi:10.3390/ijms22189982_

Round 1

Reviewer 1 Report

//

In this manuscript, Dr. Hudec and colleagues present results of a study that focuses on characteristics of the monocytic dendritic cell (MoDC) population known to play a critical role in the development of coeliac disease (CD) in humans (huCD).  In general, huCD occurs in humans expressing HLA DQ2/DQ8 and various non-HLA antigens , plus exposure to  environmental antigens….all features that strongly point to involvement of autoimmunity.  Patients present with multiple symptoms and pathologies, including extra-intestinal manifestations.  At the cellular level, upregulated and activated CD11c/CD14+ MoDCs drive Ag-specific immunity  involving the auto-Ag transglutaminase and gluten peptides generated through protein digestion.

While considerable information concerning CD has emerged in recent years, there is much to be learned concerning intervention therapies.  In this regard, the present manuscript addresses this latter issue, describing an ex vivo method that generates MoDCs from patients with CD (and/or additional pathologies), thereby permitting the basic characterization of the CD-derived MoDC population(s). Results indicate such ex vivo-derived cells express, when stimulated with LPS/gluten-derived peptides, an apparently upregulation in the common markers usedto  identify CD MoDCs.  The authors propose that this technique will be an excellent model for future molecular studies.

Critique:

Overall, this is a nicely-written manuscript with results based on fairly well-defined protocols.  However, there are several áreas of concern:

  1. The human subjects. The authors state that the subject population has been divided into three cohorts.  It is not clearly stated if the 7 controls are family members or not.  This is confusing since the authors state ¨Three Czech families (F - A, F - B, F - C) with a history of autoimmune disease participated in this study along with healthy controls from the same country¨.  Nevertheless, there does not appear to be age-matched controls for the major age groups defined by the CD families.
  2. The Institutional Review Committee´s protocol is not listed.
  3. The description presented for Figure 2 does not describe accurately the data presented. For example, were are the data for the TH group?  This raises a question of how many groups are there in the study and how many subjects are in really in the groups being studied.
  4. Supplementary figure 1 clearly shows a difference in the % of monocytes in the three age groups, as would be expcted. Without an age-matched grooup for the CD patients (who are all relatively young), how does one determine if the data are disease-associated rather than age-associated?

Author Response

Please see the attachment,

Reviewer 2 Report

The manuscript from Hudec M et al analyzed the changes in monocytes and their derivative DC in patient suffering from Celiac disease (CeD). They reported that they have identified a novel bio-profile marker that may  for a potential diagnosis of the CeD. They also elucidated the role of CD86 and HLA-DQ in MoDC as potential controller of gluten induced autoimmune response in CeD patients.

Concerns

 The major problem with this work is that it provides incrementation information on what it already known in the field of DCs in the celiac disease. Description of some data are difficult to follow.

- In the introduction, the authors completely ignored previous publications regarding the phenotype and function of DCs and monocytes in the CeD disease.

 -Cohort limitation: Number of patients (n=4) in each group in my opinion is low and therefore the conclusion might not be solid.

- Figure 2. It is not clear why the authors are analyzing  the ability of monocytes to differentiate into MoDCs following activation? It is more appropriate to analyze the proportion of MoDC before activation with LPS/PTG. FACS Data of healthy controls are not shown.

- The authors mentioned that there were no differences in the proportions of classical, intermediate and non-classical monocytes and that MoDCs were generated without prior separation of each of these subsets. I think the main question here is which of these monocyte subsets are the major sources of DCs and whether those DCs are endowed with high expression of CD86 and HAL-DQ and whether they are more inflammatory DCs (DC1) and have higher potential to activated T lymphocytes of sick patients.

- Figure 4 and 5.  data obtained in the Figures 4 and 5 are not novel and are  expected (see for example  Journal of Clinical Immunology volume 29, pages 29–37 (2009). The intracellular and cell surface expression could be easily done by FACS. A better characterization of MoDC for CeD patients and Healthy controls is missing. Similar comment on MoDCs function (cytokines production and antigen presentation)

- Why , when the authors looked at CD33 and CD65 expression on MoDCs after gating on CD14++ monocytes, however, when they generate MoDcs they used all monocytes.is

- Supplementary fig2.  The authors should show non activated MoDC derived after 6 days of culture not only activated MoDCs ( how MoDC were activated in the figure).

- FACS data suppl fig.3, the author should show also MFI of levels of HLA-DQ and CD86

- Figure 1 D. Table summarizing the sex not  the gender

Round 2

Reviewer 1 Report

The changes introduced by the authors within this revised manuscript address the main weaknesses of the original manuscript.  Necessary corrections have been made to this revised manuscript, and along with the new information is now easier to understand and follow the progress of the study.  While the data and results are interesting, more intense follow-up studies are clearly needed to confirm these prelimimary observations. 

Reviewer 2 Report

I am satisfied with the authors responses. I don't have additional comments.